# New Oxazolidinones for Tuberculosis: Are Novel Treatments on the Horizon?

**DOI:** 10.3390/pharmaceutics16060818

**Published:** 2024-06-17

**Authors:** Ricky Hao Chen, Andrew Burke, Jin-Gun Cho, Jan-Willem Alffenaar, Lina Davies Forsman

**Affiliations:** 1Sydney Pharmacy School, Faculty of Medicine and Health, The University of Sydney, Sydney, NSW 2050, Australia; rche0228@uni.sydney.edu.au; 2Department of Pharmacy, Westmead Hospital, Sydney, NSW 2145, Australia; 3Sydney Institute for Infectious Diseases, The University of Sydney, Sydney, NSW 2145, Australia; lina.davies.forsman@ki.se; 4University of Queensland Centre for Clinical Research, Faculty of Medicine, The University of Queensland, Brisbane, QLD 4006, Australia; andrew.burke@health.qld.gov.au; 5The Prince Charles Hospital, Brisbane, QLD 4032, Australia; 6Department of Respiratory and Sleep Medicine, Westmead Hospital, Sydney, NSW 2145, Australia; jin-gun.cho@health.nsw.gov.au; 7Sydney Medical School, Faculty of Medicine and Health, The University of Sydney, Sydney, NSW 2050, Australia; 8Department of Infectious Diseases, Karolinska University Hospital, SE-171 76 Stockholm, Sweden; 9Department of Medicine, Division of Infectious Diseases, Karolinska Institutet Solna, SE-171 76 Stockholm, Sweden

**Keywords:** tuberculosis treatment, linezolid, delpazolid, sutezolid, tedizolid, eperezolid, radezolid, contezolid, posizolid, TBI-223

## Abstract

Multidrug-resistant tuberculosis (MDR-TB) is a global health concern. Standard treatment involves the use of linezolid, a repurposed oxazolidinone. It is associated with severe adverse effects, including myelosuppression and mitochondrial toxicity. As such, it is imperative to identify novel alternatives that are better tolerated but equally or more effective. Therefore, this review aims to identify and explore the novel alternative oxazolidinones to potentially replace linezolid in the management of TB. The keywords tuberculosis and oxazolidinones were searched in PubMed to identify eligible compounds. The individual drug compounds were then searched with the term tuberculosis to identify the relevant in vitro, in vivo and clinical studies. The search identified sutezolid, tedizolid, delpazolid, eperezolid, radezolid, contezolid, posizolid and TBI-223, in addition to linezolid. An additional search resulted in 32 preclinical and 21 clinical studies. All novel oxazolidinones except posizolid and eperezolid resulted in positive preclinical outcomes. Sutezolid and delpazolid completed early phase 2 clinical studies with better safety and equal or superior efficacy. Linezolid is expected to continue as the mainstay therapy, with renewed interest in drug monitoring. Sutezolid, tedizolid, delpazolid and TBI-223 displayed promising preliminary results. Further clinical studies would be required to assess the safety profiles and optimize the dosing regimens.

## 1. Introduction

The treatment of multidrug-resistant tuberculosis (MDR-TB) and extensively drug-resistant tuberculosis (XDR-TB) remains a global challenge due to the limited number of effective and tolerable drugs [1].

Linezolid, originally developed to combat Gram-positive bacterial infections, has been repurposed for the treatment of tuberculosis (TB). Its mechanism of action involves the inhibition of bacterial protein synthesis by binding to the 23S ribosomal RNA of the 50S subunit, thereby impeding bacterial growth. After proving to be highly active in vitro [2,3,4] and in vivo [5,6], the first clinical studies confirmed that this drug was a promising addition to the available treatment options.

The binding of linezolid to ribosomal RNA is not selective for bacteria, and its effect on mammalian mitochondria results in the inhibition of mitochondrial protein synthesis, leading to mitochondrial dysfunction [7]. Linezolid-associated adverse effects, including lactic acidosis, hematological toxicity and peripheral neuropathy, are of concern, especially during prolonged TB treatment, and necessitates careful monitoring and management of adverse effects [8]. These severe adverse effects can often alter the course of treatment, resulting in treatment interruptions that may compromise the overall efficacy of linezolid.

Despite causing significant adverse effects, linezolid has been included in both standard longer and shorter regimens for M/XDR-TB [9]. Given the narrow therapeutic window, the optimal treatment duration, dose and safety are the focus of ongoing research. As with other drug classes, the severity of adverse effects may differ between compounds within a drug class. Due to the severe adverse effects of linezolid, several other compounds within the oxazolidinone class have been explored against TB to find a more tolerable but equally or more effective alternative. We hypothesize these novel oxazolidinones to have the potential to be incorporated into routine TB treatment, offering superior safety and efficacy compared to linezolid.

Finding the balance between efficacy and toxicity is important to guide the recommendations as to which oxazolidinone can be utilized for prolonged treatment. The ratio of mitochondrial protein synthesis inhibition to the minimal inhibitory concentration (MIC) against *M. tuberculosis* has been proposed to select patients with a positive benefit/risk ratio [10]. Mitochondrial protein synthesis inhibition values can be generated using mitochondrial biogenesis inhibition assays, which measure the effects of different drug concentrations on mitochondrial cyclooxygenase 1and the cytoplasmic synthetized succinate dehydrogenase-A proteins. An IC50 value is generated, with a higher number correlating with less toxicity. When expressed as a ratio with the in vitro antimicrobial efficacy (MIC), a result is produced which allows for a comparison between drugs with a higher number, suggesting a more favorable toxicity–benefit profile [10].

The aim of this review is to explore the oxazolidinone landscape for TB and determine if and when we can substitute linezolid with a more tolerable and efficacious alterative.

## 2. Materials and Methods

A search was performed in PubMed (Bethesda, MD, USA) using the keywords TB and oxazolidinone to identify compounds from the oxazolidinone class that were eligible for investigation. The identified compounds and synonymous drug names were subsequently used as keywords together with TB to provide a more targeted search to retrieve information on in vitro models using dynamic drug concentrations, in vivo studies and human trials.

An additional search was performed using the ClinicalTrials.gov database (Bethesda, MD, USA) to identify any studies investigating an oxazolidinone for TB which had not yet been published.

Only dynamic in vitro models were considered, as they offer an advantage over experiments using static drug concentrations [11]. In these dynamic models, the drug concentration–time profiles mimic human drug exposure, which allows for a more rapid translation of data into dose regimens that are suitable for testing in clinical trials [12,13].

The following data were extracted for all included studies: model/study type, TB strain, bacterial load, oxazolidinone dose and treatment duration. Data relating to the treatment outcome (bacterial CFU reduction) and PK results (AUC and Cmax) were extrapolated from in vitro, in vivo and human studies. In vitro studies also included MIC and AUC/MIC data. Clinical studies also assessed the mean change in time to detection (MGIT) per day.

## 3. Results

The search using TB and oxazolidinone resulted in 1582 hits in Pubmed, identifying the following oxazolidinones in addition to linezolid: sutezolid, tedizolid, delpazolid, eperezolid, radezolid, contezolid, posizolid and TBI-223 (see Figure 1). The subsequent search for these compounds resulted in 32 preclinical and 12 clinical studies, while 9 additional clinical trials were identified via ClinicalTrials.gov.

Results on oxazolidinones that were included in this review are presented in Table 1 (in vitro studies), Table 2 (in vivo and human studies), Table 3 (clinical studies) and Table 4 (pharmacokinetic data in healthy subjects).

### 3.1. Sutezolid

Sutezolid, previously known as PNU-100480, is a thiomorpholine oxazolidinone and structural analogue of linezolid (Figure 1). Sutezolid differs from linezolid in having a sulfur atom instead of an oxygen atom in the ring structure (Figure 1) [14]. It undergoes extensive first-pass metabolism to sulfoxide (U-101603), which then undergoes further renal excretion and demonstrates a plasma binding level of 48% [5,15]. CYP3A4 and flavin containing monooxygenases contribute 20% to 30% of its metabolism [16]. The concentration of the sulfoxide metabolite is up to seven times that of the parent compound; however, it performs less intracellular killing of *M. tuberculosis* than the parent compound [16]. A study of 23 TB isolates of varying susceptibility in the BACTEC MGIT 960 system showed sutezolid displaying MICs that were three times lower than that of linezolid [17] (MIC range ≤0.0625 to 0.5 mg/L). The mitochondrial protein synthesis IC50/MIC50 for sutezolid is higher than other oxazolidinones, raising the possibility of favorable clinical outcomes [10].

#### 3.1.1. In Vitro Studies

The available hollow fiber modeling studies were all published as conference abstracts and were limited to studies using *M. tuberculosis* reference strain H37Rv. These studies suggested that sutezolid had superior activity to linezolid [18,19]. A comparison of the activity of sutezolid and its metabolite U-101603 demonstrated that the latter had greater activity against nonreplicating persisters than its parent compound, which was more active in the log growth phase [20]. The addition of rifampicin was synergistic with sutezolid and was needed to prevent resistance associated with sutezolid monotherapy [21]. In monotherapy, resistance is seen at doses of less than 3000 mg/day, with the time above the MIC being the index that is most associated with efficacy [22]. This differs from linezolid, in which the AUC/MIC is more predictive of efficacy [23].

Novel in vitro models using whole-blood activity culture (WBA) for intracellular TB showed that sutezolid in combination with bedaquiline had superior activity, approaching that of rifampicin and isoniazid [24]. This study confirmed a time-dependent action, with no increase in sutezolid activity when concentrations increased above 1 mg/L. There was antagonism when low (0.1 mg/L) or medium (1 mg/L) concentrations of sutezolid were combined with low (0.1 mg/L) concentrations of pretomanid [24].

#### 3.1.2. In Vivo Studies

Many of the preclinical studies have been directed at determining if sutezolid can be used with standard first-line agents to shorten the treatment duration or with other novel agents to develop a pan-TB regimen that can treat both drug susceptible TB and MDR-TB. A murine model of TB showed that when sutezolid was added to rifampicin, isoniazid and pyrazinamide, there was a lower relapse rate of 4 vs. 2 months for sutezolid add-on therapy (5% vs. 35%) [25,26]. Conversely, the addition of linezolid was antagonistic, possibly due to its effect of reducing the absorption of isoniazid and pyrazinamide. In non-human primate studies, both sutezolid and its metabolite showed linear pharmacokinetics with first-order elimination [27]. In a mouse model, sutezolid combined with bedaquiline and pretomanid had superior sterilizing activity compared to first-line antibiotics in susceptible TB, with the addition of sutezolid being required for superiority [28]. In a murine model mimicking latent TB infection, sutezolid was significantly more potent than linezolid [29]. Another latent TB mouse model suggested that sutezolid combined with bedaquiline and pretomanid was likely to be an effective regimen [30]. Both drugs were bacteriostatic against actively growing bacilli but bactericidal against nonreplicating cells. Given their similar MICs and a three-fold higher steady-state AUC than linezolid, it is suggested that sutezolid may have other more favorable properties. These include a potentially higher permeability in eukaryotic cell membranes, with higher intracellular accumulation into macrophages [29].

#### 3.1.3. Clinical Studies

A pharmacokinetic/pharmacodynamic analysis of 50 TB subjects in a phase 2 study of sutezolid was performed using whole-blood bactericidal activity as the outcome parameter [16]. The optimum bactericidal activity based on the modeling of patient plasma concentrations was with a divided dose of 600 mg twice daily compared to 1200 mg daily, which indicates T > MIC as the relevant PK/PD index. Contrary to hollow fiber studies, sutezolid accounted for more of the intracellular killing than its metabolite PNU-101603.

An early bactericidal activity in a sputum study compared 600 mg twice daily and 1200 mg daily of sutezolid with the first-line TB treatment (HRZE) over 14 days in patients with TB and showed that sutezolid was safe, with bactericidal activity in both sputum and blood [31]. The median MIC for sutezolid was ≤0.062 mg/L (BACTEC MGIT 960). Unlike the comparator arm of HRZE, there was little EBA in the first 2 days of treatment, suggesting that sutezolid has little activity against extracellular TB. This extracellular population is considered important for resistance development but less so for relapse. There were similar EBA and WBA effects between daily and split dosing; however, the drug exposure was superior with twice daily dosing, with a greater AUC for 600 mg twice daily.

The TB Alliance tested a new tablet formulation of sutezolid in healthy volunteers as a single-ascending-dose double-blind RCT in dose ranges from 300 mg to 1800 mg [15]. There were no significant adverse events. In the fasting state, there was a dose-proportional increase in AUC, but not in Cmax. This suggested that as the dose increases, there may be saturation of absorption in the upper gastrointestinal tract, with delayed completion in the lower gastrointestinal tract.

There are several phase 2 studies that have completed recruitment or are recruiting. The PanACEA study in smear-positive drug-sensitive TB is a sutezolid dose-finding study in combination with bedaquiline, delamanid and moxifloxacin (NCT03959566). An extension of this study is the PanACEA -STEP2C -01, which compares high-dose rifampicin arms with different combinations, with one arm including sutezolid 1200 mg (NCT05807399). A 4-month phase 2c trial of a pan-TB regimen will test two different doses (1200 mg and 1600 mg once daily) of sutezolid with bedaquiline and pretomanid (NCT05686356).

### 3.2. Tedizolid

Tedizolid (previously known as torezolid, TR-700 or DA-7157) is a second-generation oxazolidinone which has a novel D-ring structure and a hydroxymethyl C-5 side chain (Figure 1) compared with linezolid, resulting in increased activity against linezolid-resistant organisms [32,33,34]. The oral phosphorylated prodrug, tedizolid phosphate (tradename Sivextro; previously TR-701, DA-7218), improves bioavailability by increasing its aqueous solubility and is rapidly converted to its active moiety by endogenous phosphatases [35]. Tedizolid phosphate (manufactured by Cubist Pharmaceuticals) has been approved in the USA for treatment of acute bacterial and skin structure infections since 2014 [36].

Tedizolid in vitro studies using broth microdilution assays have demonstrated MIC values for *M. tuberculosis* between 0.125 and 0.5 μg/mL for drug-susceptible and drug-resistant (including MDR) TB strains [37,38,39,40,41,42,43], while the MIC values for linezolid-resistant TB strains are between 1 and 16.0 [42].

#### 3.2.1. In Vitro Studies

Three hollow fiber TB infection model studies have been performed using tedizolid [40,41,44]. In a 28-day HFS model of intracellular TB (seen in disseminated TB in children), tedizolid did not inhibit electron transport chain enzyme genes or kill monocytes, which demonstrates the potentially improved safety profile compared with linezolid [40]. In addition, the tedizolid EC80 killed 4.05 log CFU/mL at 28 days compared with the control and killed 4.0 log CFU/mL higher than linezolid EC80 [40]. In an adult-type model of TB, a 42-day HFS study demonstrated a powerful sterilizing effect of tedizolid monotherapy against extracellular semidormant TB bacteria. Monte Carlo experiments showed that at an oral dose of 200 mg/day, EC80 was achieved in 92% of patients at a susceptibility breakpoint of MIC 0.5 mg/L and was not associated with mitochondrial toxicity [41]. Srivastava et al. [44] performed three HFS studies using tedizolid in combination with moxifloxacin and faropenem. Firstly, bactericidal activity was demonstrated in log-phase growth bacilli using a combination of tedizolid 200 mg and high-dose moxifloxacin at 800 mg daily, which showed no bacterial growth after 14 days of treatment. Secondly, semidormant bacilli under acidic conditions (pH 5.8) were effectively sterilized with daily tedizolid 200 mg/moxifloxacin 800 mg after 42 days, and finally, a three-drug combination of tedizolid 200 mg daily, moxifloxacin 800 mg and faropenem twice daily in dormant bacilli (nonreplicating persisters associated with latent TB infection) resulted in negative culture after 14 days of treatment; in comparison, a standard regimen of daily isoniazid 300 mg/rifampicin 600 mg/pyrazinamide 1.5 g achieved the same outcome after 21 days.

#### 3.2.2. In Vivo Studies

The pharmacokinetics of tedizolid were assessed in BALB/c mice models, displaying an overall proportional 2-fold increase in exposure (AUC) and maximum concentration (Cmax), correlating with the dose increase from 10 to 20 mg/kg. Tedizolid at 10 mg/kg was evaluated as an alternative to linezolid and sutezolid when used in combination with bedaquiline and pretomanid. At this dose, the efficacy within the first 2 months of treatment were less favorable for tedizolid, resulting in higher lung CFU counts and a larger proportional positive-culture relapse in the mice when compared to linezolid at 50 mg/kg. Additionally, the higher lung CFU counts demonstrate a lack of early bactericidal activity at 1–2 months of treatment compared to other novel oxazolidinones [28]. Alternatively, drug diffusion into the alveolar epithelial lining fluid has been examined in pneumonia-based murine models, with higher doses of tedizolid (20 mg/kg) displaying extensive penetration when compared to linezolid [45].

#### 3.2.3. Clinical Studies

In healthy subjects, both oral and IV tedizolid have a half-life of approximately 11 h [46], allowing for once-daily dosing. The absolute bioavailability following oral tedizolid is 91.5% [47]; therefore, dose adjustments are not required when changing between oral and IV routes. Tedizolid phosphate, the prodrug, has a short half-life of approximately 10 min due to its rapid conversion to tedizolid by phosphatases [47]. The AUC and Cmax increase linearly with daily doses between 200 and 600 mg [46]. Food does not affect the AUC or plasma half-life, but Tmax is delayed by 6 h compared with the fasting state, and Cmax decreases by ~26% [46]. A pooled analysis of seven clinical studies suggests that dose adjustments are not required for age, sex, race, body weight, BMI or renal and hepatic function, despite a modest increase in the tedizolid AUC by up to 34% in severe hepatic impairment [48].

There have been no clinical studies using tedizolid to date for TB; however, a 7-day phase 2 trial comparing the bactericidal activity of tedizolid versus (1) linezolid and (2) standard quadruple therapy is planned (NCT05534750).

### 3.3. Delpazolid

Delpazolid (LCB01-0371) is a novel oxazolidinone synthesized by LegoChem BioSciences Inc. (Daejeon, Republic of Korea) with a cyclic amidrazone (Figure 1), allowing for slow drug accumulation and good excretion, which may reduce long-term adverse effects [49]. Initial studies demonstrated good in vitro and in vivo activity against Gram-positive bacteria [50]. Food slightly delays Tmax but does not produce any significant differences in the AUC; therefore, the oral form may be taken with or without food [51]. A study of 120 MDR-TB isolates and 120 XDR-TB isolates showed that delpazolid’s MIC90 was 0.5 and 1.0 μg/mL, respectively. For MDR-TB isolates, delpazolid’s MIC90 (0.5 μg/mL) was lower than linezolid’s MIC90 (1.0 μg/mL) with more linezolid-resistant isolates than delpazolid-resistant isolates; however, there were no significant difference in resistance rates for XDR-TB isolates when comparing delpazolid and linezolid, despite a higher MIC90 for delpazolid (1.0 μg/mL) versus linezolid (0.25 μg/mL) in the microplate alamar blue assay [52].

#### 3.3.1. In Vitro Studies

No hollow fiber models have been reported for delpazolid.

#### 3.3.2. In Vivo Studies

No in vivo studies have been performed for delpazolid.

#### 3.3.3. Clinical Studies

The first phase 1 single-ascending-dose trial used doses between 50 and 3200 mg in healthy male subjects and demonstrated that daily doses up to 2400 mg were well tolerated, with only mild adverse events of nausea, dizziness and headache; however, moderate gastrointestinal adverse effects occurred at 3200 mg [53]. A 7-day phase 1b multiple-ascending-dose study with doses between 400 mg and 1600 mg twice daily showed good tolerability up to 1200 mg with only mild adverse events [54]. A 21-day phase 1b trial of 800 or 1200 mg twice daily showed no evidence of myelosuppression but there was a higher incidence of adverse events at 1200 mg dosing, especially diarrhea and nausea. However, no severe adverse events occurred at all trialed doses [55].

Delpazolid is rapidly absorbed following oral ingestion, with the Cmax occurring within one hour after dosing [53,55]. The bioavailability of oral delpazolid at a dose of 800 mg is close to 100% of the intravenous form, allowing for a switch between IV and oral forms without requirements for a loading dose or adjustment in dosing [56]. Delpazolid is rapidly cleared, with a t1/2 value of 1.64 h at a 800 mg oral dose [56], leading to little accumulation over time, a feature which may be important in minimizing toxicity during long-term treatment [49].

A phase 2a randomized controlled trial examining early bactericidal activity with delpazolid monotherapy over 14 days demonstrated activity in adults with non-MDR pulmonary TB, with a reduction in the sputum log-CFU which was approximately 22.9% (800 mg daily), 27.6% (400 mg twice daily) and 22.4% (800 mg twice daily) as a percentage of the CFU reduction in a control group receiving HRZE [57]. Delpazolid’s bactericidal activity was greater than linezolid monotherapy at doses of 300 mg daily or twice daily [58] but less than that seen with linezolid monotherapy at 600 mg twice daily, which had a 80.2% reduction in log-CFU of the HRZE group [57]. In view of its promising toxicity profile, a phase 2b RCT using delpazolid at varying doses in combination with bedaquiline, delamanid and moxifloxacin is currently underway in smear-positive drug-sensitive pulmonary TB subjects (NCT04550832) [59].

### 3.4. Eperezolid

Eperezolid (U-100592) is the first oxazolidinone compound made by Pharmacia & Upjohn Inc. (Kalamazoo, MI, USA), who subsequently also developed linezolid and sutezolid and published in vitro results for both antibiotics in 1996 [60]. Eperezolid and linezolid, both piperazine oxazolidinones, were developed based on the structure of an earlier oxazolidinone, DuP 721, which had been discontinued from further study due to toxicity problems in animal trials [60]. Eperezolid differs from linezolid by having a 4-(hydroxyacetyl) piperazine-1-yl group, while linezolid has a morpholin-1-yl group (Figure 1). Similar to linezolid, eperezolid inhibits bacterial protein synthesis by binding to the 50S ribosomal subunit, thereby inhibiting 70S initiation complex formation [61]. An eperezolid broth microdilution assay demonstrated potent in vitro activity against drug-susceptible *M. tuberculosis* strains with MIC values of 0.125 to 0.5 [60,62], and MICs between 0.5 and 2 were seen in multiple-resistant TB strains [62].

#### 3.4.1. In Vitro Studies

No hollow fiber studies have been performed for eperezolid.

#### 3.4.2. In Vivo Studies

Human studies demonstrated a dose-proportional linear increase in eperezolid’s plasma steady-state AUC and Cmax values at single oral doses between 50 and 1000 mg, and it was well tolerated [63]. However, further research involving eperezolid was not conducted beyond these phase 1 studies after linezolid was selected of the two candidates for further testing [64] based on its superior pharmacokinetic profile [65]. Despite promising in vitro study results, eperezolid had little activity against TB in a murine model at a dose of 100 mg/kg, with only a minor reduction in TB organisms in the lung and spleen at 4 weeks from controls, compared with much greater activity seen in INH- and linezolid-treated mice [5].

#### 3.4.3. Clinical Studies

To our knowledge, no clinical studies have been performed or are being planned at this stage.

### 3.5. Contezolid

Contezolid was developed for the treatment of multidrug-resistant (MDR) Gram-positive bacterial infections, such as methicillin-resistant Staphylococcus aureus (MRSA). It was approved in China in 2021 for the treatment of complicated skin and soft tissue infections [66]. Contezolid, also a protein synthesis inhibitor, was specifically designed to reduce the risk of myelosuppression and monoamine oxidase (MAO) inhibition. The drugs exhibit lower MAO inhibition compared to linezolid due to its altered chemical structure (Figure 1) [67]. The oxygen of the morpholine ring is excluded, which accelerates metabolism and reduces toxicity. Moreover, the presence of a 2,3-dihydropyridin-4-one (DHPO) ring in contezolid leads to decreased MOA inhibition and fewer clinically adverse drug–drug interactions by metabolization, mainly by other pathways than the p450 [68].

The in vitro activity of contezolid was similar to that of linezolid against both drug-susceptible and drug-resistant *M. tuberculosis* (7H10 broth microdilution assay, MIC50 0.5 mg/L and MIC90 1 mg/L) [69].

#### 3.5.1. In Vitro Studies

No hollow fiber studies have been performed for contezolid.

#### 3.5.2. In Vivo Studies

In a murine model, contezolid demonstrated comparable bactericidal activity to linezolid and significantly reduced the *M. tuberculosis* bacterial load in the lungs of mice. In the study, 47 mice were infected intranasally with *M. tuberculosis* Erdman and randomized to different dose regimens (50–100 mg/kg), including an untreated group. The reduction in bacterial load was comparable to linezolid, with the highest activity seen with doses of 100 mg/kg linezolid or contezolid once daily (3.61 ± 0.42 log colony-forming units (CFUs) ± SD versus 3.76 ± 0.47 log CFUs ± SD, respectively) [69].

Recently, the efficacy of contezolid was shown to be similar to linezolid when evaluated in a TB mouse model and combined with bedaquiline and pretomanid [70]. At doses of 100 mg/kg for both drugs, the mean CFU counts at 4 weeks were comparable for both linezolid and contezolid (0.52 log CFUs ± SD and 0.81 log CFUs ± SD, respectively). Similar exposures were also identified after a single oral dose of contezolid (90 mg/kg) and linezolid (100 mg/kg) in mice, resulting in a mean AUC of 200.8 μg·h/mL and 243.5 μg·h/mL, respectively [69].

#### 3.5.3. Clinical Studies

Contezolid is rapidly absorbed, and drug exposure is improved if the drug is taken with food. The maximum plasma concentration is seen at a median of 2.5 h, with non-linear pharmacokinetics at doses exceeding 600 mg. No dose adjustments are needed despite mild renal and hepatic impairment. The efficacy of contezolid for the treatment of Gram-positive bacterial infections in humans has been evaluated in several studies. Contezolid was found to have a favorable safety profile with no serious myelosuppression or neurotoxicity [66]. No significant QT prolongation was seen in a study with healthy volunteers (*n* = 52) [71]. In a phase 2 study, contezolid was found to be as effective as linezolid against complicated skin infections, with a lower frequency of hematological suppression [72].

Contezolid was used in 25 patients treated for MDR/RR-TB in China, who had adverse events to linezolid. The linezolid-related adverse events subsided or improved in 90% of cases [73]. A case report described the concentrations of contezolid in the cerebrospinal fluid (CSF), as part of a multidrug regimen against central nervous system (CNS)-TB. A 34-year-old female was treated for drug-susceptible TB meningitis, where linezolid replaced ethambutol due to visual disturbances. Linezolid was thereafter replaced by contezolid at 800 mg twice daily due to myelosuppression and symptoms of polyneuropathy. The concentrations of contezolid in the CSF were similar to the estimated unbound fraction of contezolid in serum (10%). The treatment continued for 7 months, with reversal of numbness symptoms and normalization of the white blood count, as well as clinical improvement [74]. To the best of our knowledge, no clinical studies in TB patients have been performed or are being planned at this stage.

### 3.6. Posizolid

Posizolid (AZD5847, previously AZD2563) was developed by AstraZeneca for the treatment of Gram-positive bacterial infections. Based on the mechanism of action of other oxazolidinones, posizolid was proposed to act by binding to the mycobacterial 50S ribosome subunit of the 23S rRNA, inhibiting normal intracellular protein synthesis [75]. Posizolid has a chemical structure of (5R)-3-(4-{1-[(2S)-2,3-dihydroxypropanoyl]-1,2,3,6-tetrahydropyridin-4-yl}-3,5-difluorophenyl)-5-[(isoxazol-3-yloxy)methyl]-1,3-oxazolan-2-one (Figure 1). The overall isoxazole compound differs from linezolid in that it contains a dihydroxypropanoyl group and an aryl-tetrahydropyridine substituent that are responsible for the higher potency and better antibacterial activity in both in vivo and in vitro models [76].

The MIC range for posizolid was 0.125 to 4 for drug-susceptible and single-drug-resistant *M. tuberculosis* as per the BACTEC MGIT 960 system. Additionally, the MIC range for multidrug-resistant and extensively drug-resistant strains were 0.5 to 2 and 0.5 to 4, respectively. The MIC50 and MIC90 were 1 mg/L for all strains of *M. tuberculosis* [77]. In a Wayne and Haynes model of hypoxia, greater bactericidal activity was demonstrated against nonreplicating *M. tuberculosis,* resulting in a 1.25 log CFU reduction at concentrations > 10mg/L, indicating potential sterilizing activity [29].

#### 3.6.1. In Vitro Studies

No hollow fiber studies have been performed for posizolid.

#### 3.6.2. In Vivo Studies

Increased bactericidal activity, measured as the lung CFU count in mice, was seen when posizolid (125 mg/kg) was added to pretomanid and bedaquiline, resulting in a 3.67 log CFU ± SD and 6.31 log CFU ± SD reduction at 28 and 56 days, respectively. However, the activity was less pronounced than for the other studied oxazolidinones such as linezolid and sutezolid [28]. Furthermore, a subsequent in vivo experiment in mice showed no activity of posizolid against dormant, nonreplicating bacilli [29].

An alternative study in BALB/c mice determined posizolid to be efficacious in both acute and chronic infection models. A 1.0 log CFU was observed in the chronic infection model at a once daily dose of 256 mg/kg or 128 mg/kg for two or four weeks, respectively. The drug displays linear pharmacokinetics with a direct correlated increase in the AUC and dose after oral administration. Additional analysis on drug exposure determined an AUC in the epithelium lining fluid to be two-fold greater than the plasma. This suggests significant drug penetration and highlights the significant potency against contained, stationary-growth bacilli in granulomas [78]. Moreover, by deterministic simulations, the authors concluded that posizolid at the studied dosages resulted in less than the minimum PK-PD threshold required for bactericidal activity in a mouse model [78] and was less favorable than linezolid and sutezolid [79].

#### 3.6.3. Clinical Studies

The antimycobacterial activity of posizolid was studied in a phase II study, where 60 patients with drug-susceptible TB were randomized to receive four different dose regimens for 2 weeks (500 mg once daily, 1200 mg, once daily, 500 mg twice daily or 800 mg twice daily). Only a modest bactericidal activity of posizolid was seen, measured as changes in the sputum CFU count and time to positivity in BACTEC MGIT, and it was less pronounced than that of linezolid. Hematologic and hepatic adverse events were frequent, with two serious events requiring hospitalization of the participant (thrombocytopenia and hyperbilirubinemia) [80].

As with the pharmacokinetic data from the aforementioned study, a saturable absorption was noted, with a plateau at 800 mg, indicating that higher doses might not be feasible to administer [79].

In 2016, AstraZeneca announced that they have discontinued posizolid from further development, and the planned phase 2 studies were cancelled [81].

### 3.7. TBI-223

TBI-223 is a new, orally available oxazolidinone developed by the Global Alliance for TB Drug Development and has not yet been assigned a generic name. Instead of the morpholine ring structure and amide group in linezolid, it has a 2-oxa-6-azaspiro [3.3]heptane and methyl carbamate fragments to enhance its hydrophobicity and pharmacokinetics [82]. The drug blocks protein synthesis by inhibition of rRNA (i.e., the binding of N-formylmethionyl-tRNA to the ribosome) [83]. TBI-223 has shown good oral bioavailability and a short half-life in mice [84] and proven activity against Gram-positive bacteria, including MRSA, in mouse models with different sites of infections (blood, skin and bone) [85]. Bone marrow depression was less pronounced than that of linezolid when studied in animal models [86,87].

#### 3.7.1. In Vitro Studies

There are no hollow fiber or in vitro studies for *M. tuberculosis.*

#### 3.7.2. In Vivo Studies

A mice experiment was performed to assess the potential of TBI-223 to replace linezolid in a multidrug regimen against TB disease. The addition of TBI-23 enhanced the bactericidal and sterilizing activity of the regimen. No significant difference in bactericidal activity or relapse rates in mice was seen between TBI-223 and linezolid when combined with bedaquiline and pretomanid for 4 weeks, although TBI-23 exhibited a smaller reduction by 0.5 log CFU of *M. tuberculosis* compared to that of linezolid. The authors concluded that TBI-223 is at least as efficacious as linezolid and may be a safer alternative to linezolid for the treatment of TB [88].

#### 3.7.3. Clinical Studies

The Global Alliance for TB Drug Development has evaluated the safety, tolerability and pharmacokinetic profile in healthy subjects in two completed but not yet published phase 1 studies (NCT04865536 and NCT03758612) [88].
pharmaceutics-16-00818-t001_Table 1Table 1Description of in vitro studies using hollow fiber, agar dilution and cell viability assay infection models.First Author Drug NameModel TypeTB Strain (n)Bacterial Load (log_10_ CFU/mL)Dose(mg/kg/d)Intervention Duration (weeks)CFU Reduction Outcome (log_10_ CFU/mL/d)AUC (μg·h/mL)MIC (μg/mL)AUC/MICRuiz [39]TedizolidBactec MGIT 960PS-TB (36)N/A<0.5 ^a^N/AN/AN/AMIC_50_ 0.25 (0.125–0.5) ^b^MIC_90_ 0.5(0.125–0.5) ^b^N/ASDR-TB (59)MIC_50_ 0.25 (0.06–0.5) ^b^MIC_90_ 0.5(0.06–0.5) ^b^MDR-TB (25)MIC_50_ 0.25 (0.125–0.5) ^b^MIC_90_ 0.5 (0.125–0.5) ^b^Wang [43]TedizolidMABAH37RvN/A0.0625–64 ^a,b^N/AN/AN/A0.25N/ADS-TB (17), MDR-TB (52)MIC_50_ 0.125MIC_90_ 0.25ECOFF 0.25Deshpande [40]TedizolidBactec MGIT 960H37Ra, H37Rv, CDC 1551, HN 878N/AN/AN/AN/AN/A0.25N/ABactec MGIT 960SS18bN/AN/AN/AN/AN/A0.125N/AHollow fiberH37Ra5.360–8 ^a,b^40.14 ^c^0–139.41 ^b^0.5EC_80_ 188.7Srivastava [41]TedizolidBactec MGIT 960H37RvN/AN/AN/AN/AN/A0.25N/AHollow fiber200 ^d^
60.1731.0 ± 6.60.25EC_80_ 200 Srivastava [44]TedizolidHollow fiberH37Rv(log-phase growth)3.0TZD 200 mg ^d,e^210.28 ± 0.12N/A0.25N/AH37Rv(slowly replicating bacilli at pH 5.8)TZD 200 mg ^d,e^420.17 ± 0.027N/A0.25N/ASS18b(nonreplicating persisters)TZD 200 mg ^d,e,f^560.53 ± 0.09N/A0.25N/AAono [42]TedizolidBactec MGIT 960MDR-TB (54) strains lineage 2.2.1 (Beijing)5.00.015–16 ^a,b^N/AN/AN/AMIC_50_ 0.25 (0.125–0.50) ^b^MIC_90_ 0.5(0.125–0.50) ^b^N/AH37Rv (7) strains resistant to LZDN/AN/AN/AN/AN/A1.0 -16.0 ^b^N/AVera-Cabrera [37]TedizolidMABASDR-TB (9), 25 MDR-TB (25), PS-TB (61) 5.0–5.7N/AN/AN/AN/AMIC_50_ 0.25 (0.125–0.5) ^b^MIC_90_ 0.5 (0.125–0.5) ^b^N/AMolina-Torres [38]TedizolidMiddlebrook 7H10 AgarH37Rv5.091 ^a^ (1 × MIC) 72 h1.2 ^c^N/A1N/A16 ^a^ (16 × MIC)N/A1.3 ^c^N/A1N/AZurenko [62]EperezolidMiddlebrook 7H10 AgarDS-TB (5)N/A0.03–2 ^a,b^3N/AN/A0.125–0.5 ^b^N/AMDR-TB (5)N/AN/A0.5–2 ^b^N/ABrickner [60]EperezolidMiddlebrook 7H10 AgarH37RvN/A0.03–2 ^a,b^3N/AN/A≤0.125N/AZong [52]DelpazolidMABAMDR-TB (120)N/AN/AN/AN/AN/AECOFF 2.0MIC_50_ 0.5 (<0.016–4) ^b^MIC_90_ 0.5 (<0.016–4) ^b^N/AXDR-TB (120)ECOFF 2.0MIC_50_ 0.5 (<0.016–>16)MIC_90_ 1.0 (<0.016–>16)Shoen [69]ContezolidMiddlebrook 7H10 AgarH37Rv, ATCC 35,801 (strain Erdman)5N/AN/AN/AN/AMIC_50_ 0.5MIC_90_ 1.0 N/AWallis [24]SutezolidBactec MGIT 960H37RvN/A0.1 ^a^N/A0.442 N/AN/AN/A1 ^a^0.6642 ^a^0.642TB, tuberculosis; n, number; CFU, colony-forming unit; Cmax, maximum serum concentration; AUC, area under the curve; MIC, minimum inhibitory concentration; PS-TB, pan-susceptible tuberculosis; SDR-TB, single-drug-resistant tuberculosis; MDR-TB, multidrug-resistant tuberculosis; MABA, microplate alamar blue assay; MIC_50_, lowest concentration required to inhibit 50% of isolates; MIC_90_, lowest concentration required to inhibit 90% of isolates; ECOFF, epidemiological cut-off value; EC_80_, 80% of effective maximal concentration; N/A, not applicable; *Mycobacterium tuberculosis* (M. tb) is considered drug-susceptible unless otherwise stated; results are presented as mean ± SD unless otherwise specified; ^a^, concentration administered (µg/mL); ^b^, range; ^c^, compared with control; ^d^, fixed dose; ^e^, moxifloxacin (800 mg/day), given in combination; ^f^, faropenem twice daily, given in combination.
pharmaceutics-16-00818-t002_Table 2Table 2Description of in vivo and human studies.First Author Drug NameModel TypeTB StrainInoculation MethodBacterial Load (log_10_ CFU/Lung ± SD)Route of AdministrationDose (mg)Dose Frequency (h)Treatment Duration (weeks)PK dataAbsolute CFU Reduction (log_10_ CFU/mL)CFU Reduction per Week Cmax(μg/mL)AUC(μg·h/mL)Flanagan [46]TedizolidHealthy subjectsN/AN/AN/APO20024Single dose2.0 ± 0.425.4 ± 4.6N/AN/A4003.8 ± 1.056.1 ± 13.26005.2 ± 0.779.3 ± 31.38005.5 ± 1.291.8 ± 12.912009.5 ± 1.9123.1 ± 31.220024Day 151.8 ± 0.422.5 ± 6.5300Day 212.7 ± 0.531.2 ± 6.6400Day 214.7 ± 0.552.0 ± 5.1Flanagan [47]TedizolidHealthy subjectsN/AN/AN/AIV100N/ASingle dose1.2 ± 0.217.4 ± 1.8N/AN/A2002.6 ± 0.632.6 ± 8.33004.5 ± 1.151.9 ± 11.24005.1 ± 0.858.7 ± 11.6Kim [27]TedizolidNon-human primatesN/AN/AN/APO3 ^d^248 days1.46 ± 0.5124.9 ± 7.7N/AN/ATasneen [28]TedizolidBALB/c miceN/AN/AN/APO10 ^d^N/ASingle dose6.4 ± 0.640.4 ± 2.9N/AN/A20 ^d^11.4 ± 1.677.7 ± 6.1Tasneen [28]TedizolidBALB/c miceH37RvInhalation exposure system6.17 ± 0.27PO10 ^d,e^24 ^l^8N/AN/AWk4: 1.97 ± 0.130.493Wk8: 4.50 ± 0.410.562Kim [89]TedizolidHealthy subjectsN/AN/AN/APO200N/ASingle dose2.7 ± 0.530.0 ± 6.6N/AN/A4005.1 ± 1.959.8 ± 12.86007.0 ± 1.978.9 ± 17.5IV2003.1 ± 0.531.5 ± 6.8Chen [90]TedizolidHealthy subjectsN/AN/AN/APO200N/ASingle dose2.25 (1.66 –3.23) ^b^26.1 (18.2–33.1) ^b^N/AN/A2002412.36 (1.72–3.42) ^b^25.1 (18.8–30.6) ^b^IV200Single dose3.02 (1.86–3.89) ^b^30.5 (22.9–40.2) ^b^2003 days3.49 (2.95–4.41) ^b^N/ACho [54]DelpazolidHealthy subjectsN/AN/AN/APO40012Single dose5.62 ± 2.397.79 ± 2.96 ^m^N/AN/A15.11 ± 1.988.25 ± 2.77 ^m^800Single dose11.15 ± 5.8819.48 ± 4.43 ^m^114.17 ± 4.6928.08 ± 7.77 ^m^1200Single dose13.83 ± 2.0738.11 ± 14.19 ^m^120.25 ± 7.1541.12 ± 12.76 ^m^1600Single dose26.75 ± 13.7873.44 ± 34.95 ^m^Cho [56]DelpazolidHealthy subjectsN/AN/AN/AIV200 mg/60 minN/ASingle dose2.92 ± 0.465.63 ± 1.0N/AN/A400 mg/60 min5.25 ± 0.969.42 ±1.46800 mg/60 min12.16 ± 2.4925.81 ± 8.12800 mg/30 min16.69 ± 4.0523.16 ± 5.211200 mg/30 min20.31 ± 3.2228.82 ± 2.67PO800 mg8.20 ± 3.4718.85 ± 4.98Zurenko [64]EperezolidN/AN/AN/AN/APO1000614.25 days784026600N/AN/ACynamon [5]EperezolidCD1 MiceATCC 35,801 (strain Erdman)Intravenous through caudal vein6.85 ^a^PO100 ^d^24 ^l^4N/AN/A0.92 ^p^ ± 0.110.23 ^p^Kim [57]DelpazolidSSPTB (+) patientsN/AN/A1PO800242N/AN/A0.044 ± 0.0160.022400120.053 ± 0.0170.027800120.043 ± 0.0160.0221200240.019 ± 0.0170.001Li [88]TBI-223BALB/c miceH37RvInhalation exposure system8.85 ± 0.15PO100 ^d,j^244N/A1793.39 ± 0.400.84887.23 ± 0.540.9047.94 ± 0.27100 ^d,k^46.10 ± 0.631.53Zhang [29]PosizolidBALB/c miceH37Rv, SS18bInhalation exposure system7.0 ^a^PO125 ^d^244N/AN/AN/AN/ATasneen [28]PosizolidBALB/c miceN/AN/AN/APO50 ^d^N/ASingle dose20.43 ± 2.0874.76 ± 6.14N/AN/A200 ^d^N/ASingle dose31.37 ± 1.27220.25 ± 34.95N/AN/AAlsultan [79]PosizolidPatients with DS-TBN/AN/AN/APO50024N/AN/AN/AN/AN/A120024N/A5001212680012201Furin [80]PosizolidPatients with DS-TBN/AN/A6.10 (5.59–6.61) ^c^PO5002425.56 (4.58–7.00) ^n^
43.97 (39.81–50.71) ^n^0.04 ^p^0.02 ^p^5.77 (5.09–6.45) ^c^12002428.40 (7.82–10.05) ^n^73.89 (56.88–83.12) ^n^0.070.0356.31 (5.67–6.96) ^c^5001227.69 (7.32–9.17) ^n^64.75 (54.12–70.32) ^n^0.540.276.28 (5.62–6.94) ^c^80012211.54 (10.13–12.05) ^n^93.19 (79.81–105.36) ^n^0.230.115Kim [57]SutezolidNon-human primatesN/AN/AN/APO20 ^d^244 days7.71 ± 4.9528.28 ± 18.04N/AN/A40 ^d^244 days1.2 ± 1.242.28 ± 1.81N/AN/A40 ^d^123 daysN/AN/AN/AN/ATasneen [28]SutezolidBALB/c miceN/AN/AN/APO50 ^d^N/ASingle dose1.29 ± 7.073.36 ± 4.99N/AN/ALanoix [91]SutezolidC3HeB/FeJ miceH37RvInhalation exposure system6.7–7.7 ^a,b^PO50 ^d^244N/AN/A0N/ABALB/c miceH37RvInhalation exposure system6.2–6.8 ^a,b^PO50 ^d^244N/AN/A2.130.533Zhang [29]SutezolidBALB/c miceH37Rv, SS18bInhalation exposure system7.0 ^a^PO100 ^d^244N/AN/A0.30.075Lanoix [30]SutezolidBALB/c miceH37RvInhalation exposure system4.23 ± 0.05PO50 ^d^248N/AN/AWk4: 0.9 ± 0.220.225Wk8: 1.64 ± 0.520.205C3HeB/FeJ miceH37Rv4.32 ± 0.41PO50 ^d^248N/AN/AWk4: 0.85 ± 0.400.213Wk8: 1.72 ± 0.430.215Williams [92]SutezolidBALB/c miceH37RvInhalation exposure system7.27 ± 0.44PO50 ^d,f^24 ^l^8N/AN/AWk4: 3.79 ± 0.570.948Wk8: 6.9 ± 0.750.86350 ^d,g^Wk4: 3.9 ± 0.740.975Wk8: 7.270.90950 ^d,h^Wk4: 3.28 ± 0.890.82Wk8: 6.3 ± 1.180.788Williams [26]SutezolidBALB/c miceH37RvInhalation exposure system7.92 ± 0.15PO160 ^d,i^24 ^l^8N/AN/A7.210.901Williams [25] SutezolidBALB/c miceH37RvInhalation exposure system7.49 ± 0.11PO50 ^d^24 ^l^4D1: 6.07D24: 4.32D1: 7.33D24: 8.742.19 ± 0.530.548100 ^d^24 ^l^2.4 ± 0.140.625 ^d^12 ^l^1.96 ± 0.140.4950 ^d^12 ^l^2.58 ± 0.270.645Shoen [93]SutezolidC57BL/6 miceATCC 35,801 (strain Erdman)Inhalation exposure system6.00 ^a^PO100 ^d^243 daysN/AN/A0.81 ± 0.15N/ACynamon [5]SutezolidCD1 MiceATCC 35,801 (strain Erdman)Intravenous through caudal vein6.85 ^a^PO100 ^d^24 ^l^4N/AN/A3.26 ± 0.330.815Wallis [31]SutezolidSSPTB (+) patientsN/AN/A6.88 ± 1.11PO60012N/A0.986 (36) ^o^6.494 (35) ^o^N/A0.176 ± 0.1266.92 ± 1.201200241.972 (50) ^o^7.127 (36) ^o^0.221 ± 0.080Zhu [16]SutezolidSSPTB (+) patientsN/AN/AN/APO60012N/AN/AN/AN/A0.269 (0.237–0.293)1200240.186 (0.160–0.208)Wallis [94]SutezolidHealthy subjectsN/AN/AN/APO1001220.2524 (43) ^o^0.8455 (43) ^o^N/AN/A300120.4587 (45) ^o^2.133 (23) ^o^600120.9427 (20) ^o^4.294 (23) ^o^1200242.016 (50) ^o^10.10 (30) ^o^600124N/AN/ATB, tuberculosis; CFU, colony-forming unit; SD, standard deviation; PK, pharmacokinetics; Cmax, maximum serum concentration; AUC, area under the curve; PO, oral administration; IV, intravenous administration; mg, milligrams; N/A, not applicable; wk, week; D, day; SSPTB (+), smear sputum positive pulmonary tuberculosis; results are presented as mean ± SD unless otherwise specified; *Mycobacterium tuberculosis* (M. tb) is considered drug-susceptible unless otherwise stated; ^a^, estimated amount; ^b^, range; ^c^, 95% confidence interval; ^d^, mg/kg; ^e^, bedaquiline (50 mg/kg) and pretomanid (50 mg/kg), given in combination; ^f^, bedaquiline (25 mg/kg), clofazimine (20 mg/kg) and PA-824 (50 mg/kg), given in combination; ^g^, bedaquiline (25 mg/kg) and clofazimine (20 mg/kg), given in combination; ^h^, bedaquiline (25 mg/kg) and PA-824 (50 mg/kg), given in combination; ^i^, rifampicin (10 mg/kg), isoniazid (25 mg/kg) and pyrazinamide (150 mg/kg), given in combination; ^j^, bedaquiline (25 mg/kg) and pretomanid (100 mg/kg), given in combination; ^k^, S587 (25 mg/kg) and pretomanid (100 mg/kg), given in combination; ^l^, administered for 5 days a week; ^m^, AUD_0–12_; ^n^, median (IQR); ^o^, coefficient of variation (CV%); ^p^, CFU.
pharmaceutics-16-00818-t003_Table 3Table 3Summary of clinical studies, including ClinicalTrials.gov (accessed on 7 March 2024).First AuthorDrug Name Trial PhaseStudy DesignPatients (n) Type of TB Treatment RegimenDose (mg)Duration (weeks) CFU Reduction (Log_10_ CFU/Ml) per DayMean Change in Time to Detection (MGIT) per DayAUC (mg⋅h/L)Cmax (mg/L)MIC (mg/L)AUC/MICKim [57]DelpazolidPhase 2 RCT15SSPTB (+)PO QD80020.044 (22.9%) ^a^2.69N/AN/AN/AN/A16PO BD400 0.053 (27.6%) ^a^
0.3016PO QID8000.043 (22.4%) ^a^2.3316PO QD12000.019 (9.9%) ^a^0.68Wallis [95]SutezolidPhase 1RCT19Healthy subjectsPO QD1500N/AN/AN/AN/A659 ± 165N/AN/A
10000.37 ± 0.06839 ± 386
6000.06N/A
3000.08 ± 0.07N/ARCT, randomized control trial; PO, oral administration; QD, once daily; BD, twice daily; QID, four times a day; SR, slow-release; PD, pharmacodynamics; Cmax, maximum serum concentration; AUC, area under the curve; MIC, minimum inhibitory concentration; N/A, not applicable; SSPTB (+), sputum smear positive pulmonary tuberculosis; ^a^, decline as a percentage of control group.
pharmaceutics-16-00818-t004_Table 4Table 4Description of pharmacokinetic data on healthy subjects.First Author Drug NameDose (mg)V_d_/*F* (L)CL/*F* (L/h)Ka (h^−1^)*F*Flanagan [46]Tedizolid20095.7 (23.5)6.08 (1.08)N/A0.91 (0.87–0.96) ^l^40087.0 (18.0)5.58 (1.23)600101 (23.5)6.58 (3.00)800101 (14.0)6.65 (0.94)1200116 (29.2)7.77 (2.24)200 ^b^117 (21.9)7.48 (2.12)200 ^e^108 (38.1)7.16 (1.99)300 ^b^116 (27.8)7.98 (1.80)300 ^f^127 (22.0)7.51 (1.60)400 ^b^64.8 (9.96)5.66 (0.835)400 ^f^108 (20.4)5.82 (0.607)Flanagan [47]Tedizolid100 ^h^74.5 (9.4)4.8 (0.5)N/A0.915200 ^h^67.1 (15.3)5.4 (1.8)300 ^h^61.2 (15.2)4.9 (0.9)400 ^h^67.5 (12.2)5.8 (1.1)200 ^h,i^71.5 (12.7)5.9 (1.5)200 ^i^100.1 (17.7)6.5 (1.9)Kim [89]Tedizolid20089.75.7 (1.3)N/A0.952 (0.927–0.978) ^l^200 ^h^92.15.5 (1.2)40091.35.7 (1.1)60099.06.6 (1.7)Chen [90]Tedizolid20092.4 (66.8–119) ^m^6.31 (4.98–9.06) ^m^N/A0.855 (69.3–105) ^m^200 ^h^78.3 (59.7–105) ^m^5.39 (4.09–7.19) ^m^200 ^c^98.9 (86.5–114) ^m^6.56 (5.37–8.75) ^m^Cho [54]Delpazolid400 ^c^158.84 (115.53)53.14 (17.53)N/AN/A800 ^c^72.26 (22.81)31.12 (11.65)1200 ^c^81.24 (27.38)32.88 (14.82)1600 ^b^76.7 (30.47)24.76 (7.98)Choi [55]Delpazolid800108.244.1 (18.3)N/AN/A800 ^a^86.439.9 (14.2)1200 ^a^65.130.1 (7.5)Cho [53]Delpazolid50121.18 (10.45)57.47 (4.42)N/AN/A100123.93 (31.3)62.28 (20.31)200136.36 (16.93)65.12 (11.02)400122.46 (20.10)55.16 (11.83)80097.21 (20.81)41.84 (7.80)160094.96 (42.28)31.58 (8.10)2400153.79 (79.01)30.29 (9.72)320085.57 (39.01)24.95 (6.02)Cho [56]Delpazolid^h^200 mg/60 min87.48 (11.89)36.43 (6.23)N/A0.998 (0.206)400 mg/60 min92.48 (13.57)43.31 (6.5)800 mg/60 min76.39 (9.93)33.17 (8.47)800 mg/30 min83.1 (13.67)35.84 (7.03)1200 mg/30 min96.5 (18.06)41.96 (3.9)Bulitta [96]Contezolid80017.116N/A0.640Yang [97]Contezolid1000 ^h^67.773.05 (0.89)N/AN/A1500 ^h^43.092.24 (0.62)2000 ^h^57.582.83 (1.03)15002.970.13 (0.03)1500 ^a^N/A0.16 (0.04)Li [98]Contezolid6000.530.16 (0.04)1.48 (0.19)N/A8000.660.18 (0.04)1.52 (0.31)Eckburg [99]Contezolid400 ^k^61.0 (35.1)28.3 (12.1)N/AN/A400 ^j^29.0 (5.5)16.3 (3.5)800 ^k^86.1 (32.5)35.2 (13.7)800 ^j^31.2 (7.7)16.7 (5.1)1200 ^k^150 (102)36.5 (10.0)1200 ^j^45.1 (9.9)20.6 (5.1)800 ^a^23.4 (4.5)13.9 (2.6)800 ^a,d^32.4 (17.1)13.2 (1.6)800 ^a^24.7 (7.5)14.6 (4.3)800 ^a,g^35.3 (16.5)14.3 (6.1)Wu [100]Contezolid80024.5 (13.1)8.83 (2.29)1.51 (0.70)N/A120022.8 (15.1)6.19 (1.62)2.19 (0.38)160026.9 (10.7)6.98 (1.57)2.54 (0.89)Bruinenberg [15]Sutezolid300990 (335)167 (38.2)N/AN/A6001360 (436)161 (27.1)12002060 (898)167 (54.8)18002040 (382)145 (55.4)V_d_/F, apparent volume of distribution; CL/F, apparent clearance; F, bioavailability; N/A, not applicable; results are presented as mean ± SD unless otherwise specified; doses are presented as a single oral dose unless otherwise specified; ^a^, twice daily; ^b^, day 1; ^c^, day 7; ^d^, day 14; ^e^, day 15; ^f^, day 21; ^g^, day 28; ^h^, intravenous administration; ^i^, crossover study; ^j^, fed status; ^k^, fasting status; ^l^, 90% CI; ^m^, mean (range); linezolid pharmacokinetic data are presented as follows: V_d_/F (L): 40–50, CL/F (L/h): 6–12, F.

### 3.8. Landscape Analysis

Data on new oxazolidinones highlight a scarcity of results from in vitro studies, especially hollow fiber models, which have only been conducted for tedizolid. Other studies consist mostly of susceptibility tests using microdilution assays. In vivo studies are better represented, with all oxazolidinones demonstrating activity in murine TB models. Clinical studies vary across different drugs, with only the earlier discovered oxazolidinones—sutezolid, tedizolid and delpazolid—having undergone or being in the process of recruiting patients for phase 2 clinical trials (Figure 2).

Currently, most novel oxazolidinones are in the process of undergoing phase 1 or 2 clinical trial studies to evaluate the drug safety, pharmacokinetics and treatment response. Eperezolid and contezolid have not been subject to any planned or completed clinical studies at this stage. Posizolid had phase 2a trials completed; however, its discontinuation by AstraZeneca in 2016 has resulted in the cancellation of subsequent planned phase 2 trials (Figure 3) [81].

TB-based clinical trials are a time-consuming process which involves a systematic process of planning and executing. Patient recruitment after protocol submission and ethical approvals is a lengthy process involving multiple considerations such as patient eligibility, trial site and study/sample size. Slow recruitment rates can often delay trial completion, resulting in increased costs [101]. Studies evaluating M/XDR-TB treatments are often longer than for drug-susceptible TB, resulting in treatment durations ranging from 18 to 24 months and patient follow-ups post-treatment can often vary from 6 months (primary outcome) to 30 months (secondary outcomes), resulting in total trial durations of up to several years [102]. Phase 2 clinical trials are often conducted to assess drug safety and efficacy, whereas phase 3 trials involve larger and diverse patient populations to evaluate the safety, efficacy and adverse effects. Trial data from these studies are utilized to support the New Drug Applications (NDAs) submission to the relevant regulatory agencies such as the Food and Drug Administration (FDA) and European Medicines Agency (EMA) for comprehensive review, ultimately leading to approval for use in the treatment of TB [101].

Available funding for TB-based clinical trials has historically been scarce and comparatively less than in other fields of medicine like oncology and cardiology. However, with the contemporary emergence of M/XDR-TB strains, there has been a gradual increase in available funding for TB research, with USD 1 billion in 2021 [103]. Despite the increased support, available funds still fall short of the WHO’s TB funding target. A recent phase 3 drug-resistant TB trial (TB-PRACTECAL) by the Médecins Sans Frontières (MSF) on linezolid demonstrated total costs of up to EUR 34 million. In line with increasing costs, MSF proposed a project to promote increased transparency and publication of the costs of clinical trials to facilitate the development of new international funding policies [104]. Sufficient funding would promote more clinical trials to be conducted in conjunction and fast-track the development of novel TB drugs to market and clinical use.

Phase 2 clinical trials are a critical point in dictating the progression of a drug, especially if outcomes in terms safety or efficacy are compromised. Eperezolid was originally developed in the late 1990s in conjunction with linezolid to target MDR Gram-positive bacterial infections. After undergoing phase 1 clinical trials, there has not been any additional activity or registration for phase 2/3 trials, suggesting that the development of the drug has ceased. Similarly, posizolid was originally developed as a broad-spectrum Gram-positive antibiotic, which was determined to also display anti-TB activity in the early 2000s. After undergoing phase 2a trials, which resulted in unfavorable safety outcomes, it was discontinued in 2016.

With the substantial rise in the emergence and prevalence of M/XDR-TB strains in TB patients, first-line anti-TB drugs such as isoniazid and rifampicin become ineffective in treating the infection. In 2012, the novel antimycobacterial bedaquiline was able obtain a fast-tracked approval by the FDA via the accelerated approval program, permitting it to overcome the lengthy process of phase 3 clinical trials before it could be used. This was formulated on the basis of its completed phase 2b clinical trials, showing favorable results on safety and efficacy [105]. As such, this provisional approval grants patients with urgent life-threatening MDR-TB earlier access to lifesaving and promising treatments, preventing further disease progression or resistance. Similarly in 2019, another anti-TB drug (pretomanid) obtained the same accelerated approval for use in the treatment of M/XDR-TB, in combination with bedaquiline, linezolid and moxifloxacin, as part of the BPaLM regimen [101]. Novel oxazolidinones can attempt to follow similar approaches to obtain accelerated approval; however, as bedaquiline and pretomanid are already indicated to target resistant strains, this may reduce the urgency of the approval of new drugs for the treatment of M/XDR-TB.

## 4. Discussion

The value of linezolid as a core treatment option for MDR-TB has sparked interest in this class of antimicrobials. However, because of the toxicity of linezolid, there is a clear need for a safer and more tolerable alternative. Various oxazolidinone antibiotics are currently being evaluated in in vitro, in vivo and clinical studies for TB. Sutezolid, tedizolid, delpazolid, contezolid and TBI-223 show promising preclinical data, but the clinical evaluation of these compounds has been limited to early-phase clinical trials for tedizolid and TBI-223 and phase 2 trials for sutezolid and delpazolid.

There have been delays in the development of novel anti-TB agents due to high rates of failure in clinical trials, resulting in concerns regarding safety and adverse effects. Drug discovery/development is often a lengthy and expensive process, and the reliance on me-too drugs, whereby slight structural modifications are made to existing drug molecules to provide a slight pharmacotherapeutic advantage, are not considered long-term solutions for the treatment of M/XDR-TB. However, repurposing existing drugs may provide an alternative solution for streamlining the drug development process at reduced costs. These drugs can form the foundation for novel lead compounds or be utilized to fast-track preclinical dose-finding studies [106].

Of the oxazolidinones currently under investigation, tedizolid, delpazolid, eperezolid, contezolid and posizolid are repurposed drugs, while TBI-223 and sutezolid have been developed specifically for TB from origin.

Sutezolid, synthesized at the same time as linezolid, was only investigated more recently, exhibiting superior activity against TB in in vitro models. Its metabolism and toxicity profile are also different and may help to overcome the toxicity experienced with linezolid.

Tedizolid, a second-generation oxazolidinone repurposed for TB, demonstrates potent activity against TB in both in vitro and in vivo studies. Pharmacokinetic studies show promising results for once-daily dosing, which is helpful for programmatic TB treatment. The drug appears to be well tolerated without any exposure-response-related gastrointestinal adverse effects.

Within the TB-based clinical studies for newer oxazolidinones, no adverse effects were reported for sutezolid, tedizolid, eperezolid or contezolid. Delpazolid was discovered to cause mild nausea/vomiting and moderate gastrointestinal adverse effects at higher doses, whereas posizolid phase 2 clinical trials resulted in a few patients experiencing severe thrombocytopenia and hyperbilirubinemia.

Delpazolid exhibits comparable activity to linezolid against *M. tuberculosis* in vitro. Clinical studies demonstrate good tolerability and efficacy in reducing the sputum bacterial load in TB patients. Eperezolid and posizolid both showed promising in vitro activity; however, its efficacy in animal models of TB and early-phase clinical trials compared to the current standard linezolid was inferior. Hence, further clinical development is not expected for these two drugs. Contezolid exhibits activity against drug-susceptible and drug-resistant TB strains in vitro. In murine models, it demonstrates comparable bactericidal activity to linezolid. Clinical studies in humans suggest favorable safety profiles. TBI-223, a newly designed oxazolidinone, showed potential as an alternative to linezolid for TB treatment in preclinical studies, with ongoing phase 1 trials to evaluate its safety and pharmacokinetics.

Previous studies have identified specific structural conformations within oxazolidinones to be responsible for either antibacterial activity or toxicity. The fluorine on the phenyl ring was identified to be accountable for antibacterial activity, whereas the functional groups within the C-5 position were determined as the key precipitant for mitochondrial toxicity and MAO-A inhibition [107,108]. Specifically, the S configurations at the C-5 position were identified in all oxazolidinones and are responsible for mitochondrial protein synthesis inhibition and additionally associated with antibacterial activity. When comparing linezolid with other oxazolidinones, the thiomorpholine structure was a weaker mitochondrial protein synthesis inhibitor than the piperazinyl group in eperezolid. Additional groups such as the 1,2,4-triazolyl and dihydropyranyl oxazolidinones were also highly inhibitive for mitochondrial protein synthesis. However, by adding a methyl substituent to the distal position of the C-ring, mitochondrial protein synthesis inhibition could be minimized without a significant impact on the antibacterial potency. C-5 substituents that contained the traditional acetamide group had minimal MAO-B effects, but greatly inhibited MAO-A [107,109]. More importantly, the presence of a morpholino or piperazinyl C-ring was the key characteristic responsible for the balance between high potency and tolerability [107].

Furthermore, drug interactions have yet to be established in clinical trials. Despite this, adverse drug interactions can be anticipated through specific structural and functional group configurations at the C-5 position. As previously mentioned, the inhibition of both MAO isoforms has been identified. As MAO-B is responsible for the breakdown of serotonin, oxazolidinones are plausible candidates for pharmacodynamic interactions with other serotonin-increasing drugs (e.g., selective serotonin reuptake inhibitors and tricyclic antidepressants), potentially triggering serotonin toxicity. Evidently, this has been identified in linezolid, where co-administration with other serotonin-increasing drugs requires frequent monitoring. Alternatively, the inhibition of MAO-A reduces norepinephrine breakdown, resulting in increased adrenergic activity. Therefore, interactions with vasopressors and sympathomimetic agents have been proposed. Additionally, clinical studies on linezolid displayed evidence of drug interactions with clarithromycin and rifampicin. In MDR-TB patients, coadministration of a 500 mg daily dose of clarithromycin resulted in a 50% increase in both the linezolid AUC and Cmax. The proposed mechanism of reduced clearance was due to the inhibition of P-glycoprotein efflux pumps, as linezolid is not a cytochrome P450 (3A4) substrate and does not undergo hepatic metabolism [110]. Alternatively, the pre-treatment administration of rifampicin was shown to decrease the linezolid AUC by 30% in mice models. Its mechanism of interaction remains unclear [111]. Nonetheless, these interactions can be anticipated in these novel oxazolidinones; however, based on the few clinical studies, no clinically significant interactions have currently been observed [112].

### 4.1. Limitations

Although this review presents a generalized perspective on the potential for oxazolidinones in the treatment of TB, there are some limitations. This review was not conducted following a systematic approach with explicit eligibility or inclusion/exclusion criteria. The methodology only involved searching the keywords in one database (PubMed), without an extensive search in the gray literature or conference abstracts. However, we believe that this scoping approach was reasonable for this study due to the novel nature of these oxazolidinones, leading to only a few preclinical studies on TB and clinical trials currently being in progress or active recruitment.

### 4.2. Future Perspectives

Novel oxazolidinones display promising results for the treatment of TB, with positive preclinical results. Due to the severe adverse effects from linezolid therapy, poorer treatment outcomes have been observed in M/XDR-TB patients. Frequent hematological effects such as thrombocytopenia and anemia, along with significant nausea and vomiting, can make treatment intolerable [8]. This can often result in early treatment discontinuation, dose reduction or substitution for alternative therapies. Currently, sutezolid, delpazolid and contezolid also display a slight advantage over linezolid in early phase 1 and 2 studies in terms of safety and tolerability, resulting in less toxicity and adverse effects at similar doses. However, all oxazolidinones are still in early clinical stages, with trials focusing on drug-susceptible TB patients. Further clinical evaluation studies are required to assess their suitability for the treatment of drug-resistant strains of TB to determine their applicability in replacing linezolid in M/XDR-TB treatment regimes. Currently, the use of linezolid is still considered the optimal choice and is expected to remain the mainstay therapy for M/XDR-TB for the next 5 to 10 years. The BPaLM regimen has enabled a shorter treatment duration of 6 months from 9 to 18 months [113]. Trial results have displayed high efficacy in treating resistant strains with improved treatment success rates (89%) [9]. Nevertheless, these drugs encompass dose-/exposure-related adverse effects with bedaquiline- and moxifloxacin-triggering corrected QT interval (QTc) prolongation and linezolid causing myelosuppression and mitochondrial toxicity at extended treatment durations > 28 days [114].

Linezolid displays complex PK/PD in different patients, making it a good candidate for drug monitoring and dose optimization [114]. Current therapeutic drug monitoring (TDM) of linezolid has been implemented in routine care to measure linezolid trough concentrations (Cmin) as a method of determining the toxicity. However, future considerations should include proactive TDM for the purpose of dose optimization rather than toxicity monitoring, as it would be more beneficial in guiding dosing decisions while avoiding toxicity altogether [115]. At present, the therapeutic thresholds have not yet been identified, although previous studies have determined the relative drug exposure (AUC/MIC) to be a good indicator of drug efficacy [114]. To facilitate TDM in TB-endemic areas, a recent study to improve the accessibility with the use of a portable mobile UV spectrophotometer has been completed with promising results. Further validations in clinical trials are required [116].

## 5. Conclusions

Various oxazolidinone antibiotics, namely, sutezolid, tedizolid, delpazolid and TBI-223, show early promise as effective treatments for TB, with ongoing research aimed at optimizing dosing regimens and evaluating their safety profiles in clinical trials.

## Figures and Tables

**Figure 1 pharmaceutics-16-00818-f001:**
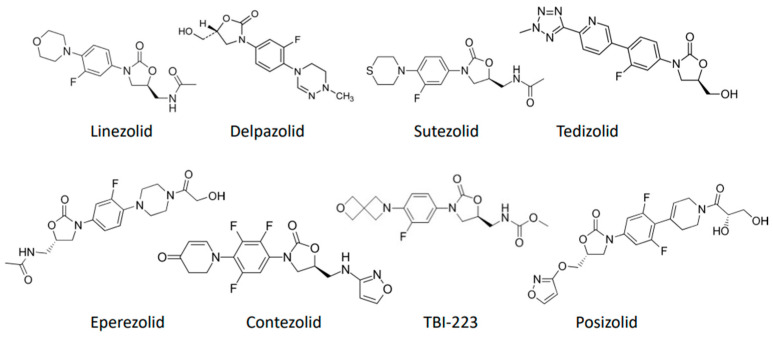
An overview of the molecular structures of the oxazolidinones investigated for TB.

**Figure 2 pharmaceutics-16-00818-f002:**
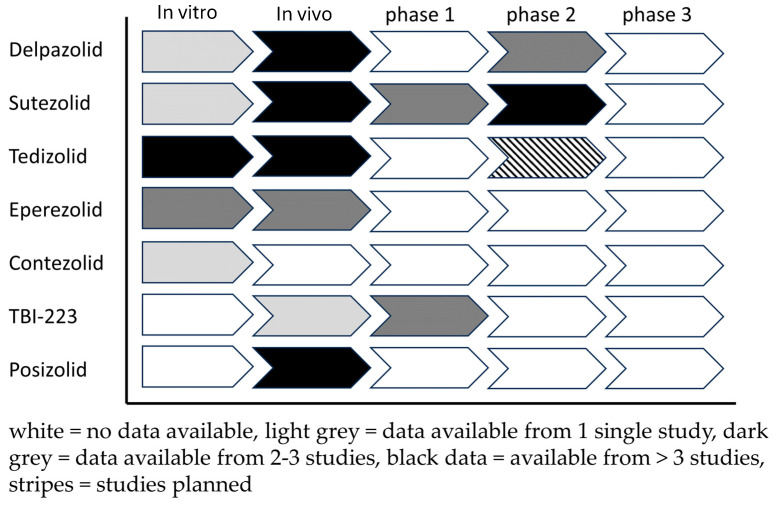
Overview of available data on oxazolidinones evaluated for TB.

**Figure 3 pharmaceutics-16-00818-f003:**
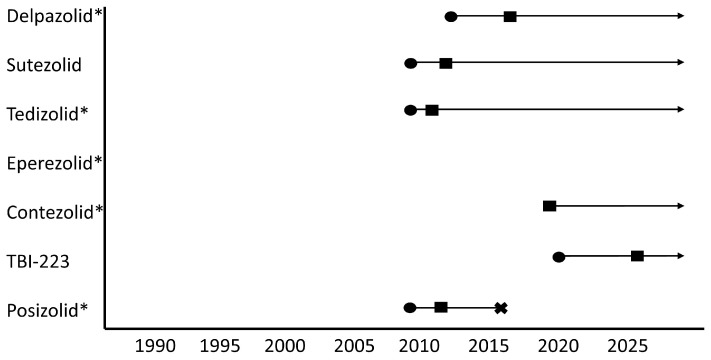
Pipeline of oxazolidinones. circles = “phase 1”, squares = “phase 2”, cross = development ceased, * = repurposed drugs.

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
