# Peer review of "New Oxazolidinones for Tuberculosis: Are Novel Treatments on the Horizon?"

_pharmaceutics, 2024, doi:10.3390/pharmaceutics16060818_

Round 1

Reviewer 1 Report

Comments and Suggestions for Authors

The review addresses a current topic, and for improvement, I propose the following:
1. In the "materials and methods" section, the introduction of a PRISMA flowchart.
2. References 18-22, 98 should comply with journal requirements.
3. References 60, 61, 62, 63 and 64 are over 20 years old and require revision, given that the topic covered is bacterial antibiotic resistance.

Author Response

.

Reviewer 2 Report

Comments and Suggestions for Authors

Dear Authors,

I have the manuscript and I send you my comments:

1) please add a table showing the difference in the mechanism of action between linezolid and these drugs

2) please add a table showing the difference in pharmacokinetic characteristics between these drugs and linezolid

3) PLease add a section on adverse drug reactions

4) please describe the potential drug interactions

5) please delete non clinical data

Comments on the Quality of English Language

none

Author Response

.

Reviewer 3 Report

Comments and Suggestions for Authors

The manuscript entitled New oxazolidinones for tuberculosis; when can we expect them?” summarizes the preclinical and clinical status of antitubercular compounds that contain oxazolidinone as a pharmacophore such as sutezolid, tedizolid, delpazolid, eperezolid, radezolid, contezolid, posizolid and TBI-223. This review aims to compare the therapeutic efficacy of linezolid with other oxazolidinone derivatives. The authors have made a good effort, however, the below revisions are recommended:

1.      Please add a suitable hypothesis. The manuscript lacks a suitable hypothesis.

2.      Please write a paragraph highlighting the significance of oxazolidinone derivatives in antitubercular therapy.

3.      I recommend splitting Table 1 into two tables: SDR-TB, single-drug resistant tuberculosis, and MDR-TB, multi-drug resistant tuberculosis. This will help the readers to get a prima facie antitubercular activity of the oxazolidinone derivatives.

4.      In 3.8. In the landscape analysis section (Fig. 2), please present lines# 549-550 in a graphic instead of text.

5.      I did not find an authentic answer to the title question (When can we expect them?). I recommend changing the title.

  1. The manuscript must be thoroughly checked, and the quality of the language must be improved. There are numerous grammatical and syntactical mistakes.
Comments on the Quality of English Language

Please see the above.

Author Response

.

Round 2

Reviewer 1 Report

Comments and Suggestions for Authors

The requested modifications or explanations have been made.

Reviewer 2 Report

Comments and Suggestions for Authors

no other comments

Comments on the Quality of English Language

none

Reviewer 3 Report

Comments and Suggestions for Authors

The authors have revised the manuscript considering my recommendations. The manuscript can be published.

Comments on the Quality of English Language

The manuscript can be published.